# Dynamic Rank Adjustment for Accurate and Efficient Neural Network Training

## Abstract

Low-rank training is a primary strategy for efficient deep learning, but it presents a fundamental challenge. It reduces computational cost, yet it permanently caps a model's representational capacity and accelerates the rank collapse that diminishes its expressive power during training. We address this with dynamic-rank training, a framework built on the intuition that a model can temporarily escape its low-rank constraints to restore its full learning potential. Our approach strategically interleaves full-rank epochs within a low-rank schedule, with the timing of these restorative phases aligned with the learning rate's noise regimes to maximize their effect. This enables the model to regain expressive power at critical stages of training by restoring the effective rank of its weights. Our extensive evaluations across various computer vision and natural language processing benchmarks show that the dynamic-rank method achieves the accuracy of full-rank models while retaining the computational advantages of low-rank training.

## 1 Introduction

Low-rank reparameterization methods have been actively studied for efficient training of large neural networks. Low-rank training strategies typically reduce the number of trainable parameters by applying matrix decompositions to a model's weight matrices. While this approach lowers training cost, it permanently caps the maximum attainable rank of those matrices, thereby limiting the model's ability to learn complex patterns. Moreover, recent studies report that the effective rank of the weights tends to decline during training (Xie et al., 2017; Huang et al., 2025). Therefore, to train large neural networks efficiently while preserving their learning capacity, it is crucial to address the decline in weight-matrix rank that occurs during low-rank training.

Singular value decomposition(SVD)-based low-rank training (Jaderberg et al., 2014) is among the most popular re-parameterization techniques. Other tensor factorizations, such as Tucker (Kim et al., 2015) and CP (Lebedev et al., 2014) decompositions, have likewise been adopted to enable low-rank training. More recently, low-rank fine-tuning methods have been proposed, including LoRA (Hu et al., 2022), AdaLoRA (Zhang et al., 2023), LoRA-GA (Wang et al., 2024), DoRA (Liu et al., 2024), and SLTrain (Han et al., 2024). Although these approaches reduce the number of trainable parameters, they often overlook the progressive decline in the effective rank of the weights, which in turn compromises learning capability. Some regularization methods, such as soft orthogonal regularizer (Xie et al., 2017) and its variants (Bansal et al., 2018; Kim & Yun, 2022), focus on tackling the rank decline issue. However, they incur higher computational cost and memory overhead, making them less practical for training large neural networks.

This study explores how to mitigate the decline in the effective rank of model weights in low-rank training. Our key finding is that interleaving a few full-rank epochs within low-rank training effectively restores the model's effective rank. Specifically, we analyze how run-time rank adjustment affects the singular value spectrum of the model weights, and we present a practical strategy for adjusting the rank during training to mitigate the decline in effective rank. Our theoretical analysis and empirical study demonstrate that dynamic rank adjustment matches the accuracy of full-rank training while retaining the system efficiency of low-rank methods.

Another key finding is that the effectiveness of rank adjustment is closely coupled with the learning rate. Our study shows that full-rank epochs should be scheduled according to the noise scale induced by the learning rate. In general, the learning rate tends to be initialized to a large value and then

Table 1: Feature-wise comparison across representative low-rank training methods.

| Low-rank Training Method | Parameter-efficient | Rank Recovery | Pre-training Compatible | Noise-scale Aware | Decomposition Agnostic |
|---|---|---|---|---|---|
| SVD-based Low-rank (Jaderberg et al., 2014) | ✓ | ✗ | ✓ | ✗ | ✗ |
| TKD-CPD (Phan et al., 2020) | ✓ | ✗ | ✓ | ✗ | ✗ |
| GKPD (Hameed et al., 2022) | ✓ | ✗ | ✓ | ✗ | ✗ |
| Soft Orthogonality (Xie et al., 2017) | ✗ | ✓ | ✓ | ✗ | ✓ |
| SRIP$^+$ (Kim & Yun, 2022) | ✗ | ✓ | ✓ | ✓ | ✓ |
| LoRA (Hu et al., 2022) | ✓ | ✗ | ✗ | ✗ | ✓ |
| AdaLoRA (Zhang et al., 2023) | ✓ | ✗ | ✗ | ✗ | ✗ |
| PELA (Guo et al., 2024) | ✓ | ✗ | ✗ | ✗ | ✓ |
| SLTrain (Han et al., 2024) | ✓ | ✗ | ✗ | ✗ | ✓ |
| SparseLoRA (Khaki et al., 2025) | ✓ | ✗ | ✗ | ✗ | ✓ |
| **Dynamic-rank (proposed)** | ✓ | ✓ | ✓ | ✓ | ✓ |

progressively decayed during training. In this study, we analyze how the learning rate affects the gap between low-rank and full-rank updates and, based on this analysis, propose a general rank scheduling framework that maximizes the benefits of interleaving full-rank epochs within low-rank training. Table 1 provides a feature-wise comparison of various low-rank training methods.

To the best of our knowledge, this is the first study to explore the benefits of interleaving full-rank training with low-rank training and to demonstrate the efficacy of a dynamic rank adjustment technique. We evaluate the proposed dynamic-rank training framework through extensive empirical studies on computer vision and natural language processing benchmarks. Furthermore, we benchmark our approach against state-of-the-art (SOTA) low-rank training techniques and regularization methods designed to restore effective rank. Across all experiments, we find that dynamic-rank training achieves accuracies comparable to full-rank training while significantly reducing computational costs, similar to conventional low-rank approaches. In addition, our results show that the proposed method integrates seamlessly with low-rank training when combined with soft-orthogonality (SO) regularization, confirming that the two techniques are complementary.

## 2 BACKGROUND

**Low-rank Re-parameterization** – Low-rank reparameterization is a model approximation technique that is popularly used in large neural network training. Given a model weight matrix $\mathbf{W}$, a matrix decomposition method is applied to $\mathbf{W}$ before training begins. For example, if SVD is used, $\mathbf{W} \in \mathbb{R}^{m \times n}$ is decomposed to two smaller matrices $\mathbf{A} \in \mathbb{R}^{m \times k}$ and $\mathbf{B} \in \mathbb{R}^{n \times k}$ such that $\mathbf{W} = \mathbf{A}\mathbf{B}^\top$, where $k < m$ and $k < n$. As $k$ decreases, the total number of model parameters is reduced, thereby lowering the computational cost of training. However, the reconstructed weight matrix can have up to $k$ ranks, resulting in the limited learning capability.

**Learning Rate and Noise Scale** – In previous works, learning rate is known to play a key role in determining generalization performance of machine learning models. It has been theoretically shown that the noise scale $g$ in gradient approximation is determined by the learning rate and batch size, as follows (Smith et al., 2017): $g \approx \eta N/B$, where $\eta$ is learning rate, $N$ is the dataset size, and $B$ is the mini-batch size. This relation links learning rate schedules to the notion of noise scale. In early training, a large noise scale helps the model explore and shape the decision boundary (Li et al., 2019; Lee et al., 2023), consistent with the common practice of decaying the learning rate. In this study, we investigate the relationship between the noise scale and the rank of model's weights.

## 3 RELATED WORK

**Low-rank Reparameterization Methods** – Several studies have explored the use of low-rank reparameterization techniques including singular value decomposition, Tucker decomposition, and CP decomposition methods (Jaderberg et al., 2014; Kim et al., 2015; Lebedev et al., 2014). Recently, GKPD proposed to re-parameterize the model weights using Kronecker-product decomposition and demonstrated promising performance (Hameed et al., 2022). PELA applies low-rank training to the pre-training phase to reduce the computational cost of training (Guo et al., 2024). FLANC uses a customized tensor decomposition method to enable direct model aggregations across clients in the context of federated learning (Mei et al., 2022). FedPara combines decomposition methods and the

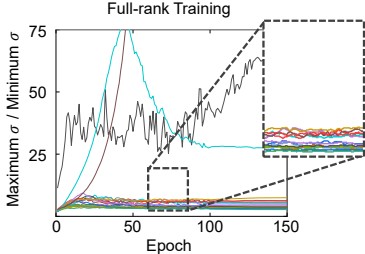 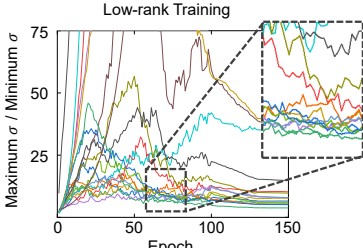

Figure 1: Comparison of the layer-wise singular-value spectral ratio ($\lambda$) across different model ranks during ResNet20 training on CIFAR-10. The left plot shows layer-wise $\lambda$ curves for full-rank training, while the right plot shows those for SVD-based low-rank training. We omit the legend since there are too many layers. Throughout the whole training, most layers in the re-parameterized model exhibit large $\lambda$ values, indicating convergence to a low-rank space.

Hadamard product to enhance the representation capacity of re-parameterized models (Hyeon-Woo et al., 2021). TKD-CPD jointly utilize Tucker and CP decompositions (Phan et al., 2020). While these methods effectively reduce the number of trainable parameters and thus the computational and communication costs, they do not consider the inherent rank decline issue.

**Low-rank Fine-tuning Methods** – Low-rank Adaptation (LoRA) (Hu et al., 2022) is one of the most popular applications of re-parameterization technique, particularly in the context of fine-tuning. The importance of singular values under LoRA's context is deeply analyzed in (Ke et al., 2024). AdaLoRA (Zhang et al., 2023) is a variant of LoRA which dynamically controls the rank of layers within a fixed total parameter budget. DoRA (Liu et al., 2024) separately fine-tunes the magnitude and the direction, based on LoRA. SLTrain (Han et al., 2024) leaves a sparse and frozen matrix besides the adaptor. LoRA-GA (Wang et al., 2024) initializes adapters using a subset of eigenvectors of gradient matrices. SparseLoRA (Khaki et al., 2025) decomposes pretrained weights using SVD.

**Rank Recovery Methods** – Some regularization methods have been proposed to address the issue of inherent rank decline. Soft orthogonality (SO) is the basic regularization method which minimizes the difference between Gram matrix of the weight matrix and the identity matrix (Xie et al., 2017). Double soft orthogonality (DSO) is a variant of SO which considers the regularization with overcomplete and undercomplete Gram matrices (Bansal et al., 2018). Spectral Restricted Isometry Property (SRIP) is a variant of RIP (Candes & Tao, 2005) that minimizes the spectral norm of the difference between the Gram matrix of a weight matrix and the identity matrix. Another study propose a sine-activated low-rank training strategy that is also designed to restore the model weight's rank (Ji et al., 2024). These methods commonly recover the effective rank of model weights, however, the rank cannot exceed the hard limit imposed by the low-rank reparameterization. In this study, we focus on how to overcome this limitation by adjusting the model rank at run-time.

## 3.1 MOTIVATION

Recently, it has been theoretically shown that the rank of gradients tends to decrease as training progresses (Huang et al., 2025; Xie et al., 2017). Consequently, given a fixed training dataset and a large number of repeated training steps, the model weights are also expected to exhibit a similar reduction in rank.

As the singular value spectrum of model weights becomes more skewed, only a few singular vectors capture most of the information, while those associated with smaller singular values contribute little to the model's representational capacity. We empirically verify this tendency by analyzing the effective rank of neural networks throughout training. To quantify how rapidly the model loses rank, we define layer-wise singular value spectral ratio, $\lambda^l$, as follows.

$$\lambda^l = \frac{\sigma_{\max}^l}{\sigma_{\min}^l}, \quad l \in [L], \tag{1}$$

where $[L]$ denotes the set of all network layers, and $\sigma_{\max}^l$ and $\sigma_{\min}^l$ are the maximum and minimum singular values of layer $l$, respectively. Assuming that $\sigma_{\max}^l$ remains reasonably small, an increase in $\lambda^l$ indicates that $\sigma_{\min}^l$ is approaching zero, meaning the model is losing its rank.

Figure 1 shows how $\lambda^l$ changes during the training of ResNet20 on CIFAR-10. On the left-side (full-rank model training), only three layers show noticeable rank reduction while the effective ranks of all other layers remain quite stable throughout the training. On the right side (reparameterized to half rank at all convolution layers), most layers exhibit much higher $\lambda$ values compared to the full-rank training, resulting in a final model with low ranks in most layers. The full rank training achieves a validation accuracy of $92.23\%$, while the low-rank reparameterized training yields $91.08\%$. Based on this observation, we conclude that low-rank model reparameterization can significantly reduce the rank of model weights, thereby harming the model's representational capacity.

Based on this empirical study, we can derive one critical insight as follows.

> *As the model is reparameterized to a lower rank,*
> *the rank of its weights tends to decrease more rapidly.*

Therefore, having full-rank epochs in the middle of training is expected to mitigate the rapid decline in the effective rank of model weights. This insight motivates the design of a general model rank adjustment framework, which we describe in the following section.

## 4 DYNAMIC-RANK TRAINING FRAMEWORK

In this section, we propose a general dynamic-rank training framework that temporarily increases the rank of the model's weight matrices at selected points during training. We first formalize rank *inflation* and *deflation*, then discuss how to schedule rank adjustments. In particular, we provide two key insights for placing full-rank epochs within low-rank training to maximize rank restoration. Figure 2 illustrates the proposed framework.

### 4.1 MODEL RANK ADJUSTMENT

We first define two directions of model rank adjustment, *inflation* and *deflation*, as follows. For simplicity, we use the notation of SVD-style low-rank reparameterization, but it can be easily replaced with other methods such as Tucker or CP decompositions.

**Model Inflation** – Low-rank training can be implemented in two different forms, vanilla low-rank reparameterization and low-rank adaptation, LoRA (Hu et al., 2022). Without loss in generality, we define model inflation process as follows. Given low-rank model weights $\mathbf{A} \in \mathbb{R}^{m \times k}$ and $\mathbf{B} \in \mathbb{R}^{n \times k}$ and base parameter $\mathbf{W}_0 \in \mathbb{R}^{m \times n}$, the model is inflated such that $\mathbf{W} \leftarrow \mathbf{W}_0 + \mathbf{AB}^\top$. Consequently, the maximum available rank is increased from $k$ to $n$. If the low-rank model follows the standard low-rank reparameterization, the initial weight matrix $\mathbf{W}_0$ is set to the zero matrix, i.e., $\mathbf{0}_{m \times n}$. Please see Appendix for the case of convolutional layers.

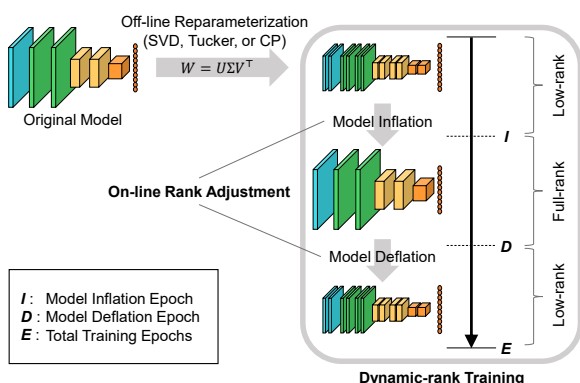

Figure 2: A schematic illustration of dynamic-rank training framework.

**Model Deflation** – Given a model weight $\mathbf{W}$, the model is deflated by attaching a low-rank adaptor path next to the original weight such that $\mathbf{W}_f + \mathbf{AB}^\top \leftarrow \mathbf{W}$, where $\mathbf{W}_f \in \mathbb{R}^{m \times n}$ is the given model weight matrix and $\mathbf{A} \in \mathbb{R}^{m \times k}$ and $\mathbf{B} \in \mathbb{R}^{n \times k}$ are low-rank model weights. The provided weight matrix is frozen as $\mathbf{W}_f$, and only $\mathbf{A}$ and $\mathbf{B}$ are trained. $\mathbf{A}$ and $\mathbf{B}$ can be initialized using either random distributions or zero matrices. Note that this naturally resembles LoRA when SVD is used as the reparameterization method.

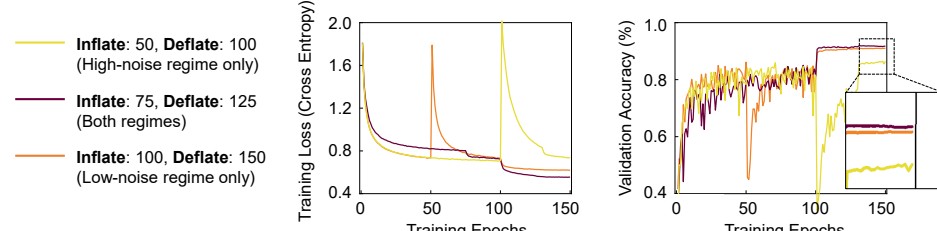

Figure 3: CIFAR-10 (ResNet20) benchmark with various dynamic-rank schedules. *Inflate* and *Deflate* indicate the epoch where the model rank is increased and decreased, respectively. The best accuracy is achieved when the full-rank epochs are located in both high-noise and low-noise regimes.

These model rank adjustment procedures incur extra computations during training, however, when the epoch budget $E$ is sufficiently large, the extra computational cost typically becomes negligible. We now turn our attention to when the model rank should be inflated or deflated during training.

## 4.2 MODEL RANK SCHEDULING

Here, we discuss how to determine appropriate timings for adjusting the model rank, taking into account the common practice of using learning rate decay for noise control in modern deep learning. Let the total training budget be $E$ epochs. Training begins in a low-rank form, with the rank of the model weights increased at epoch $I$ and reduced back at epoch $D$. That is, training is performed in the full-rank space for $D - I$ epochs and in the low-rank space for the remaining $E - (D - I)$ epochs. Under this setting, the key question is *what values of $I$ and $D$ are optimal*. We first present two practical principles for choosing $I$ and $D$ to maximize rank recovery.

**Full-rank training should be performed in both high-noise and low-noise regimes** – We propose having full-rank epochs both before and after the learning rate decay. This ensures that all ranks are sufficiently exposed to different learning rates during training. After deflating the model, only $k$ ranks are updated in the adaptor and the remaining $n - k$ ranks stay frozen until the model is inflated again. Thus, if the model is deflated before learning rate decay, the $n - k$ ranks do not get trained under the low-noise regime, potentially causing underfitting and reduced accuracy.

In contrast, if full-rank training is performed only in the low-noise regime, the model is likely to suffer from poor generalization performance. It has been shown that training in a high-noise regime during the early phase improves generalization performance (Li et al., 2019; He et al., 2019; Lee et al., 2023). Therefore, if the model is inflated for the first time in the low-noise regime, the $n - k$ ranks will be trained only with a small learning rate, potentially degrading the model's generalization performance. Moreover, scheduling the full-rank epochs later reduces the training iterations for the adapter's $k$ ranks, potentially causing underfitting.

Figure 3 presents empirical results supporting our argument. We trained ResNet20 on CIFAR-10 under three schedules, each with full-rank training epochs inserted at different points. The model is reparameterized with SVD to reduce its rank by half. Because the learning rate decays after epoch 100, the first and third schedules correspond to full-rank epochs in the high-noise and low-noise regimes, respectively. As expected, the best accuracy is achieved when full-rank epochs are located in both noise regimes. The curves spike during deflation as a randomly initialized low-rank adaptor is attached and begins training. The yellow and orange curves clearly exhibit underfitting and low accuracy. Based on this analysis, we conclude that full-rank training should be applied in both high-noise and low-noise regimes to sufficiently train the model.

**Full-rank training should be at the end of the high-noise regime and the early low-noise regime** – First, we define the effective rank of $\mathbf{W}$, denoted by $r$, as the number of singular values that are significantly greater than zero. We analyze rank bounds of the reconstructed model as follows.

**Proposition 1.** *Suppose model weight matrices: $\mathbf{W}_0 \in \mathbb{R}^{m \times n}$, $\mathbf{A} \in \mathbb{R}^{m \times k}$, and $\mathbf{B} \in \mathbb{R}^{n \times k}$ and a re-constructed matrix $\mathbf{W} = \mathbf{W}_0 + \mathbf{A}\mathbf{B}^\top$. Assuming that the effective rank of $\mathbf{W}_0$ is $r$, the rank of the reconstructed matrix $\mathbf{W}$ is bounded between $\max\{0, r - k\}$ and $\min\{m, n, r + k\}$.*

Table 2: Comparison of singular-value spectral ratio ($\lambda$) across different rank-adjustment schedules. The $I$ and $D$ indicate rank inflation and deflation, respectively. ResNet20 was trained on CIFAR-10 for 150 epochs, with the learning rate reduced by a factor of 10 at epochs 100 and 130.

| Setting | $I$ | $D$ | Acc. | $\lambda^0$ | $\lambda^{10}$ | $\lambda^{18}$ |
|---------|-----|-----|------|-------------|----------------|----------------|
| None    | -   | -   | 91.14% | $\infty$  | 7.58 | $\infty$ |
| Early   | 0   | 50  | 91.26% | 59.32     | 6.02 | 76.47 |
| Middle  | 25  | 75  | 91.82% | 39.29     | 4.27 | 74.18 |
| Late    | 50  | 100 | 91.87% | 37.51     | 4.19 | 70.36 |

*Proof.* First, re-write each matrix as a summation of rank-1 matrices such that $\mathbf{W}_0 = \sum_{j=1}^{r} \mathbf{u}_j \mathbf{v}_j^\top$ and $\mathbf{A}\mathbf{B}^\top = \sum_{j=1}^{k} \mathbf{a}_j \mathbf{b}_j^\top$. Consequently, $\mathbf{W} = \sum_{j=1}^{r} \mathbf{u}_j \mathbf{v}_j^\top + \sum_{j=1}^{k} \mathbf{a}_j \mathbf{b}_j^\top$. The minimum rank of $\mathbf{W}$ is $r - k$ if the $k$ rank-1 matrices in $\mathbf{A}\mathbf{B}^\top$ eliminate $k$ rank-1 components of $\mathbf{W}_0$. On the other hand, the maximum rank of $\mathbf{W}$ is $r + k$ if the $k$ rank-1 matrices in $\mathbf{A}\mathbf{B}^\top$ are orthogonal to all $k$ components of $\mathbf{W}_0$. Therefore, considering appropriate caps, the rank of the reconstructed matrix $\mathbf{W}$ is bounded between $\max\{0, r - k\}$ and $\min\{m, n, r + k\}$. $\qquad\square$

This analysis suggests that the rank of the reconstructed model weights is closely tied to the rank of $\mathbf{W}_0$, the frozen weights. In particular, performing full-rank training right before deflating the model maximizes $r$, enabling the model to maintain a high effective rank. The full-rank training budget, $D - I$, may be smaller than the number of epochs before the first learning rate decay. In such cases, training will naturally begin with a low-rank reparameterized model, which will then be inflated before the learning rate decay.

To connect theory and practice, we compare weight ranks under different rank schedules in Table 2. We perform CIFAR-10 (ResNet-20) training with the learning rate decaying at epoch 100. The rightmost three columns show $\lambda$ for layers 0, 10, and 18 (near the input, middle, and output). In the conventional low-rank training (*None*), $\lambda_0$ and $\lambda_{18}$ are very large, indicating widely spread spectra and effective rank loss. As full-rank training is applied later, the singular value spectra become narrow (smaller $\lambda$ values). These results demonstrate that the full-rank budget should be placed toward the end of the high-noise regime.

Now, let us discuss the impact of full-rank training in the low-noise regime. Once the learning rate decays, the rank of the model weights does not dramatically change due to the reduced magnitude of updates. Instead, what matters more is the extent to which low-rank training pushes the model away from the full-rank convergence point. The following proposition formalizes this relationship.

**Proposition 2.** *As the learning rate decays, the gap between the full-rank model update and the low-rank model update is expected to decrease.*

*Proof.* The gradient of adaptor weights, $\mathbf{A} \in \mathbb{R}^{m \times k}$ and $\mathbf{B} \in \mathbb{R}^{n \times k}$ are written as $\nabla f(\mathbf{A}\mathbf{B}^\top)\mathbf{B} \in \mathbb{R}^{m \times k}$ and $\nabla f(\mathbf{A}\mathbf{B}^\top)^\top \mathbf{A} \in \mathbb{R}^{n \times k}$, respectively. So, the reconstructed model is:

$$
\begin{aligned}
\mathbf{W}_{t+1} &= \mathbf{A}_{t+1}\mathbf{B}_{t+1}^\top \\
&= \left(\mathbf{A}_t - \eta \nabla f(\mathbf{W}_t)\mathbf{B}_t\right)\left(\mathbf{B}_t - \eta \nabla f(\mathbf{W}_t)^\top \mathbf{A}_t\right)^\top \\
&= \mathbf{A}_t\mathbf{B}_t^\top - \eta \nabla f(\mathbf{W}_t)\mathbf{B}_t\mathbf{B}_t^\top - \eta \mathbf{A}_t\mathbf{A}_t^\top \nabla f(\mathbf{W}_t) + \eta^2 \nabla f(\mathbf{W}_t)\mathbf{B}_t\mathbf{A}_t^\top \nabla f(\mathbf{W}_t).
\end{aligned}
$$

Since $\mathbf{A}_t\mathbf{B}_t^\top = \mathbf{W}_t$, the remaining three terms on the right-hand side can be considered as the low-rank update. For simplicity, we shorten the terms as follows.

$$
\mathbf{X} := \nabla f(\mathbf{W})\mathbf{B}\mathbf{B}^\top
$$

$$
\mathbf{Y} := \mathbf{A}\mathbf{A}^\top \nabla f(\mathbf{W})
$$

$$
\mathbf{Z} := \nabla f(\mathbf{W})\mathbf{B}\mathbf{A}^\top \nabla f(\mathbf{W})
$$

Then, the difference between the full-rank update $\nabla f(\mathbf{W}_t)$ and the low-rank update, $d$ becomes:

$$
d_t := \|\nabla f(\mathbf{W}_t) - \eta\left(-\mathbf{X}_t - \mathbf{Y}_t + \eta \mathbf{Z}_t\right)\|_F \tag{2}
$$

Based on triangle inequality, (2) is upper-bounded as follows.

$$
d_t \leq \|\nabla f(\mathbf{W}_t)\|_F + \eta \|\mathbf{X}_t\|_F + \eta \|\mathbf{Y}_t\|_F + \eta^2 \|\mathbf{Z}_t\|_F . \tag{3}
$$

Then, $\|\mathbf{X}_t\|_F$ on the right-hand side can be further bounded as follows.

$$\mathbf{X}_t = \|\nabla f(\mathbf{W}_t)\mathbf{B}\mathbf{B}^\top\|_F$$
$$\leq \|\nabla f(\mathbf{W}_t)\|_F \cdot \|\mathbf{B}\mathbf{B}^\top\|_2$$
$$= \|\nabla f(\mathbf{W}_t)\|_F \cdot \|\mathbf{B}\|_2^2$$

The same procedure can be applied to the $\mathbf{Y}$ and $\mathbf{Z}$ terms, yielding

$$d_t \leq \|\nabla f(\mathbf{W}_t)\|_F \left(1 + \eta\|\mathbf{A}_t\|_2^2 + \eta\|\mathbf{B}_t\|_2^2\right)$$
$$+ \eta^2\|\nabla f(\mathbf{W}_t)\|_F \left(\|\nabla f(\mathbf{W}_t)\|_2 \cdot \|\mathbf{A}_t\|_2 \cdot \|\mathbf{B}_t\|_2\right).$$

According to the bound above, the difference $d_t$ is expected to decreases as $\eta$ decays. $\qquad\square$

For clarity, we omit $\mathbf{W}_0$ from the analysis. Incorporating $\mathbf{W}_0$ into each step of the proof to recover LoRA-style low-rank training is straightforward. The above analysis shows that full-rank training should be performed as early as possible in the low-noise regime to minimize $d_t$. In general, most popular learning rate schedules monotonically reduce the learning rate over time. Therefore, scheduling full-rank epochs earlier tends to result in a smaller accumulated difference, $\sum_t d_t$, over the remaining training epochs.

### 4.3 Unified Rank Adjustment Framework

Based on the two key insights discussed above, we build up a general model rank adjustment framework as shown in Figure 2. The training begins in a low-rank space. The model's rank is adjusted twice throughout the training, increased in a high-noise regime and then decreased in a low-noise regime. In our empirical study, we observed reasonably good performance when $I$ is set to the middle of high-noise regime and $D$ to the middle of low-noise regime. For example, given $E = 150$, if learning rate decays at epoch 100, we recommend beginning with $I = 50$ and $D = 125$ and then fine-tune these values to minimize training time while maintaining model accuracy. The pseudocode is provided in Appendix A.1.

## 5 Experiments

**Experimental Settings** – We evaluate dynamic-rank training framework on three computer vision benchmarks: CIFAR-10 (Krizhevsky et al., 2009) (ResNet20), CIFAR-100 (Wide-ResNet28-10), and TinyImageNet (mnmoustafa & Ali, 2017) (ResNet50), and natural language processing (NLP) datasets in GLUE (Wang et al., 2018) (DeBERTaV3-base (He et al., 2021)) benchmark. All experiments were conducted on a GPU server with two NVIDIA RTX 4090 GPUs. Each experiment was repeated at least twice, and we report the mean accuracy with standard deviation. The detailed hyperparameter settings are presented in the Appendix.

### 5.1 Comparative Study

**Computer Vision Benchmarks** – To evaluate the performance of our dynamic-rank training strategy, we compare it with several low-rank reparameterization methods, including SVD, Tucker, and CP decompositions. We also compare it with TKD-CPD (Phan et al., 2020), a recently proposed low-rank training method that jointly utilizes Tucker and CP decompositions. Table 3 shows the results. See Appendix for $I$ and $D$ settings. The *Rank Ratio* $\rho$ indicates the ratio of the reparameterized weight rank to the original weight rank. *Comp.* refers to the average proportion of trainable parameters during training. Across all benchmarks, low-rank training leads to noticeable accuracy drops. Increasing $\rho$ to 0.75 narrows the gap but still falls short of full-rank performance. In contrast, dynamic-rank training achieves full-rank accuracy at a computational cost similar to the $\rho = 0.75$ setting. TKD-CPD dramatically reduces computation but yields the lowest accuracy among low-rank methods. Applying dynamic-rank training to TKD-CPD restores its accuracy to a level comparable to full-rank training.

**Natural Language Processing Benchmarks** – We validate dynamic-rank training framework by fine-tuning the DeBERTaV3-base on 8 datasets from GLUE benchmark (Wang et al., 2018). In fine-tuning, there is no high-noise regime, as the learning rate is set to a small value and remains in the

Table 3: Comparison of CV benchmarks across various low-rank training methods. The dynamic-rank method consistently improves accuracy while effectively reducing computational cost.

| Method | Rank Ratio $\rho$ | CIFAR-10 (ResNet20) Acc. | Comp. | CIFAR-100 (WRN28-10) Acc. | Comp. | Tiny ImageNet (ResNet50) Acc. | Comp. |
|---|---|---|---|---|---|---|---|
| Full-Rank | 1.00 | $92.15 \pm 0.1\%$ | 1.00 | $78.82 \pm 0.1\%$ | 1.00 | $60.32 \pm 0.1\%$ | 1.00 |
| SVD | 0.50 | $91.19 \pm 0.1\%$ | 0.57 | $73.57 \pm 0.1\%$ | 0.56 | $54.90 \pm 0.1\%$ | 0.79 |
| | 0.75 | $91.56 \pm 0.1\%$ | 0.84 | $76.90 \pm 0.1\%$ | 0.84 | $56.16 \pm 0.1\%$ | 0.92 |
| **Dynamic Rank (SVD)** | 0.50 | $\mathbf{92.13} \pm 0.1\%$ | 0.76 | $\mathbf{78.61} \pm 0.2\%$ | 0.64 | $\mathbf{61.80} \pm 0.1\%$ | 0.85 |
| Tucker | 0.50 | $90.27 \pm 0.1\%$ | 0.41 | $68.77 \pm 0.1\%$ | 0.39 | $60.62 \pm 0.1\%$ | 0.69 |
| | 0.75 | $91.50 \pm 0.1\%$ | 0.79 | $79.68 \pm 0.2\%$ | 0.77 | $61.07 \pm 0.7\%$ | 0.87 |
| **Dynamic Rank (Tucker)** | 0.50 | $\mathbf{92.10} \pm 0.1\%$ | 0.70 | $\mathbf{78.48} \pm 0.1\%$ | 0.66 | $\mathbf{61.25} \pm 0.1\%$ | 0.84 |
| CP | 0.50 | $88.31 \pm 0.1\%$ | 0.31 | $65.55 \pm 0.1\%$ | 0.30 | $42.05 \pm 0.1\%$ | 0.65 |
| | 0.75 | $88.90 \pm 0.1\%$ | 0.59 | $65.32 \pm 0.1\%$ | 0.57 | $43.12 \pm 0.1\%$ | 0.78 |
| **Dynamic Rank (CP)** | 0.50 | $\mathbf{91.33} \pm 0.1\%$ | 0.61 | $\mathbf{78.61} \pm 0.3\%$ | 0.65 | $\mathbf{59.56} \pm 0.5\%$ | 0.83 |
| TKD-CPD (Tucker + CP) | 0.50 | $87.99 \pm 0.1\%$ | 0.21 | $58.90 \pm 0.1\%$ | 0.18 | $34.53 \pm 0.1\%$ | 0.60 |
| | 0.75 | $89.62 \pm 0.1\%$ | 0.35 | $60.22 \pm 0.1\%$ | 0.31 | $34.84 \pm 0.1\%$ | 0.66 |
| **Dynamic Rank (TKD-CPD)** | 0.50 | $\mathbf{91.24} \pm 0.1\%$ | 0.55 | $\mathbf{73.96} \pm 0.3\%$ | 0.59 | $\mathbf{60.06} \pm 0.5\%$ | 0.80 |

Table 4: Comparison of NLP benchmark performance on GLUE (DeBERTaV3-base). The dynamic-rank method achieves the best accuracy while maintaining substantially lower computational cost.

| Method | CoLA | MNLI | MRPC | QNLI | QQP | RTE | SST-2 | STS-B | Avg | Comp. |
|---|---|---|---|---|---|---|---|---|---|---|
| Full Fine-Tuning | 0.675 | 0.9018 | 0.8995 | 0.9397 | 0.9222 | 0.8375 | 0.9587 | 0.9097 | 100% | 1.000 |
| LoRA (Hu et al., 2022) | 0.6333 | 0.8874 | 0.8799 | 0.9192 | 0.8922 | 0.7978 | 0.9461 | 0.9021 | 97.2% | **0.014** |
| AdaLoRA (Zhang et al., 2023) | 0.6529 | 0.8891 | 0.8431 | 0.9183 | 0.8891 | 0.8086 | 0.9495 | 0.8943 | 97.1% | 0.017 |
| DoRA (Liu et al., 2024) | 0.6579 | 0.8929 | 0.8774 | 0.9176 | 0.8924 | 0.7617 | 0.9506 | 0.8970 | 97.1% | 0.014 |
| SLTrain (Han et al., 2024) | 0.6652 | 0.8913 | 0.8848 | 0.9174 | 0.8934 | 0.7833 | 0.9461 | 0.8987 | 97.6% | 0.018 |
| LoRA-GA (Wang et al., 2024) | 0.6409 | 0.8899 | 0.8602 | 0.9156 | 0.8904 | 0.7617 | 0.9472 | 0.8962 | 96.4% | 0.014 |
| SparseLoRA (Khaki et al., 2025) | 0.6675 | 0.8898 | 0.8898 | 0.9148 | 0.8900 | 0.7653 | 0.9438 | 0.8960 | 97.3% | 0.014 |
| **Dynamic Rank (SVD)** | **0.6752** | **0.8959** | **0.8886** | **0.9264** | **0.9062** | **0.8267** | **0.9472** | **0.9087** | **99.0**% | 0.115 |

low-noise regime throughout training. Therefore, in these experiments, we place full-rank epochs at the beginning of fine-tuning.

We compare our dynamic-rank method with the following SOTA low-rank fine-tuning methods: LoRA, AdaLoRA, DoRA, SLTrain, LoRA-GA, and SparseLoRA. AdaLoRA follows a similar principle to our dynamic-rank method, adjusting the rank at run time. However, it requires costly SVD operations during training, whereas our method avoids explicit decomposition of the weight matrices. DoRA and LoRA-GA have exactly the same computational cost as LoRA because they focus on the initialization of adaptor weights. SparseLoRA has fewer frozen parameters than LoRA, but the number of trainable parameters is the same.

Table 4 reports the fine-tuning performance on GLUE tasks. The *Avg* column shows the relative performance with respect to full fine-tuning, averaged over the 8 tasks. All the SOTA methods substantially reduce computational cost compared to full fine-tuning. LoRA is the most efficient but suffers from a noticeable drop in accuracy. Other methods alleviate this drop, however the gap from full fine-tuning accuracy remains non-negligible. By contrast, the dynamic-rank training achieves accuracy comparable to the full fine-tuning. This comparison demonstrates that, although the dynamic-rank scheme slightly increases the computational cost compared to LoRA, it effectively restores the effective rank of the model weights, thereby improving the fine-tuning accuracy.

**Dynamic-rank Training with Regularization**

– Our dynamic-rank training strategy is readily applicable to regularization methods designed to restore the reduced effective rank. Table 5 shows CIFAR-10 benchmark results with three regularization methods, soft orthogonality (SO) (Xie et al., 2017), double soft orthogonality (DSO) (Bansal et al., 2018), and SRIP$^+$ (Kim & Yun, 2022). These regularizers effectively improve the accuracy of SVD-based low-rank training. When combined

Table 5: CIFAR-10 (ResNet20) accuracy comparison between with and without applying the proposed method to SOTA rank recovery methods.

| Setting | Low-rank | Dynamic Rank |
|---|---|---|
| Low-rank SVD (0.5) | $91.14 \pm 0.1\%$ | $\mathbf{92.09} \pm 0.1\%$ |
| Low-rank SVD (0.5) + SO | $91.61 \pm 0.1\%$ | $\mathbf{92.35} \pm 0.2\%$ |
| Low-rank SVD (0.5) + DSO | $91.76 \pm 0.4\%$ | $\mathbf{92.49} \pm 0.1\%$ |
| Low-rank SVD (0.5) + SRIP$^+$ | $91.50 \pm 0.0\%$ | $\mathbf{92.23} \pm 0.2\%$ |

with dynamic-rank training, they yield consistent accuracy gains across all regularizers. Therefore, the two are complementary, and their combination achieves high accuracy at low computational cost. However, these regularizers consume a large amount of memory space to compute Gram matrices, and the overall memory footprint may substantially increase as they are combined with dynamic-rank training.

**Parameter Heatmap Comparison**
– Figure 4 shows a parameter value comparison between low-rank and dynamic-rank trainings. We draw heatmaps with 4-D weight tensors of size $3 \times 3 \times 64 \times 64$ obtained from the largest convolution layer in ResNet20. The tensors are reshaped to $192 \times 192$ 2-D matrices. To show the difference more clearly, we reconstructed the tensor from the low-rank reparameterized weights using 50 singular vectors corresponding to the smallest singular values. The low-rank map exhibits blocky arti-

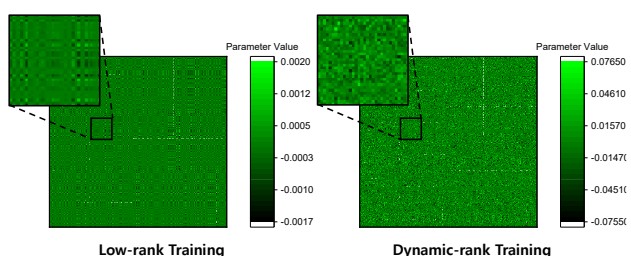

Figure 4: Parameter comparison between low-rank and dynamic-rank trainings. The heatmap shows the weights of the largest convolution layer in ResNet20 after training.

facts, whereas the dynamic-rank map does not. The presence of the artifacts implies the existence of a null space in certain directions of the feature space, meaning that variations in the input along those directions do not affect the output. This result is well aligned with the model accuracy comparisons. Therefore, we conclude that the dynamic-rank method well restores the effective rank of the model, thereby better exploiting representation capacity.

## 5.2 COMPUTATIONAL COST ANALYSIS

Here, we analyze the computational overhead of dynamic-rank training, which is shown in *Comp.* columns in Table 3 and 4. First, the full-rank training cost is calculated as

$$T_F = E \cdot d, \tag{4}$$

where $E$ is the total number of epochs and $d$ is the model size. Then, the low-rank training cost is

$$T_{SVD=0.5} = E \cdot d_{SVD=0.5}, \tag{5}$$

where the subscription $SVD = 0.5$ indicates the model is reparameterized using SVD and the rank reduction ratio $\rho = 0.5$. Finally, our dynamic-rank training cost is calculated as

$$T_{DR} = (D - I) \cdot d + (E - (D - I)) \cdot d_{SVD=0.5}. \tag{6}$$

Thus, if $\rho$ is the same, $T_{DR}$ should be higher than $T_{SVD}$ and lower than $T_F$. E.g., ResNet20 contains 272,762 trainable parameters and the reparameterized model with SVD ($\rho = 0.5$) has 155,170 parameters. When $E = 150$, $I = 60$, and $D = 135$, where $\phi = 0.5$, $T_{DR}/T_F$ becomes 0.7844, which is shown in *Comp.* column in Table 3. Our empirical study shows that a relatively small $D - I$, achieved by carefully tuning $I$ and $D$, yields accuracy comparable to full-rank training, while bringing the cost close to that of conventional low-rank training.

## 6 CONCLUSION

In this study, we propose a dynamic-rank training framework that restores the effective rank of model weights while preserving the efficiency of low-rank training. We also present two key insights for maximizing rank recovery by strategically interleaving full-rank epochs within low-rank training. Our extensive empirical study demonstrates that scheduling full-rank epochs at the end of the high-noise regime and the beginning of the low-noise regime maximizes the recovery of the model's effective rank, thereby improving accuracy. The proposed dynamic-rank training scheme is general and readily applicable to any deep learning applications. We believe that harmonizing the proposed dynamic-rank training and other compute-efficient neural network training methods can be a promising future work. Additional discussion on *potential limitations and future work* is provided in Appendix A.7.

## REPRODUCIBILITY STATEMENT

Our experimental software was developed using PyTorch 12.6. The experimental results were collected on a GPU server that contains two NVIDIA RTX 4090 GPUs. The detailed dataset information, data pre-processing, and hyper-parameter settings are provided in Appendix A.2. The model inflation and deflation steps with Tucker and CP decompositions are described in Appendix A.3. The detailed implementation of model rank adjustment at convolution layers are described in Appendix A.4. We will make our software publicly available once the paper is published.

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

---

**Algorithm 1** Dynamic-rank Training

---

**Require:** Model parameters $\Theta$, Training dataset $\mathcal{D}$, Total epochs $E$, low-rank dimension $k$, Inflation epoch $I$, Deflation epoch $D$, Learning rate schedule $\eta$.

1: Initialize model parameters $\Theta$ in a low-rank form.
2: **for** $t = 1 \rightarrow E$ **do**
3:    **if** $t = I$ **then**
4:       *// Inflate model to full-rank*
5:       **for** low-rank weight $(\mathbf{A}, \mathbf{B})$ with base $\mathbf{W}_0$ in $\Theta$ **do**
6:          $\mathbf{W} \leftarrow \mathbf{W}_0 + \mathbf{A}\mathbf{B}^\top$
7:          Replace $(\mathbf{W}_0, \mathbf{A}, \mathbf{B})$ with $\mathbf{W}$ in $\Theta$.
8:       **end for**
9:    **end if**
10:   **if** $t = D$ **then**
11:      *// Deflate model back to low-rank*
12:      **for** each full-rank weight matrix $\mathbf{W}$ in $\Theta$ **do**
13:         Freeze the current weight $\mathbf{W}_f \leftarrow \mathbf{W}$.
14:         Initialize new low-rank matrices $\mathbf{A}$, $\mathbf{B}$.
15:         Replace $\mathbf{W}$ with $(\mathbf{W}_f, \mathbf{A}, \mathbf{B})$ in $\Theta$.
16:      **end for**
17:   **end if**
18:   Train the model on dataset $\mathcal{D}$ using $\Theta$ and $\eta_t$.
19:   Update learning rate $\eta_{t+1}$ according to the schedule.
20: **end for**

---

# A  APPENDIX

The appendix is structured as follows:

- Section A.1 presents a pseudocode of the proposed dynamic-rank training framework.

- Section A.2 summarizes experimental settings corresponding to all the experimental results reported in the main manuscript.

- Section A.3 describes how we inflate and deflate the model weights using Tucker and CP decompositions.

- Section A.4 describes how we adjust the rank of model weights at convolution layers.

- Section A.5 presents a singular value spectrum ratio comparison among different model rank adjustment settings.

- Section A.6 provides the results of additional ablation study on the impact of $\phi$ on model accuracy.

- Section A.7 summarizes potential limitations of our proposed dynamic-rank training framework.

We declare that an LLM was used to polish the writing. However, its purpose was solely to improve presentation quality and check grammar.

## A.1  ALGORITHM

Algorithm 1 shows a pseudocode of the proposed dynamic-rank training framework. In this pseudocode, we assume the model is reparameterized using SVD. It is straightforward to replace SVD with other decomposition techniques. During $E$ training epochs in total, if the epoch ID $t$ becomes $I$, the model is inflated following the previously discussed reconstruction steps (line $5 \sim 8$). Likewise, if the epoch ID $t$ becomes $D$, the model is deflated (line $10 \sim 17$). Other steps are the same as general neural network training process. Therefore, it does not have any dependencies on optimizers or model architectures.

## A.2  EXPERIMENTAL SETTINGS

**CIFAR-10/CIFAR-100 datasets** – We perform the typical image preprocessing used in many previous works (Lee et al., 2023) for CIFAR-10/100 datasets. 60,000 images with 50,000 images for train

Table 6: Hyper-parameter settings for experiments shown in Table 3. The $\rho$ is the low-rank model's rank reduction ratio.

| Dataset | Batch Size | Learning Rate | Epochs ($E$) | LR Decay | Inflate/Deflate ($I, D$) | Weight decay | $\rho$ |
|---|---|---|---|---|---|---|---|
| CIFAR-10 | 128 | 0.1 | 150 | 100, 130 | 55, 120 | $1e-4$ | 0.5 |
| CIFAR-100 | | | 200 | 150, 180 | 80, 170 | $5e-4$ | |
| Tiny ImageNet | 64 | | 100 | 70, 90 | 30, 80 | $5e-4$ | |

dataset and 10,000 images for validation dataset. Each image is padded by 4 pixels on every dimension and then randomly cropped to the original size. Then, we normalize and standardize the values for all individual pixels. Finally, with probability of 0.5 we randomly flip the image horizontally.

**Tiny ImageNet dataset** – For Tiny ImageNet with 200 classes, we augment the data samples during training as follows: aspect ratio adjustment [0.8, 1.25], random resizing [256, 384] pixels on shorter side, random cropping to 224×224 then resizing to $64 \times 64$, horizontal flipping with probability of 0.5, and HSV color augmentation (hue $\pm 36$ degree, saturation/brightness [0.6, 1.4]). We normalize using ImageNet standard RGB values (mean [0.485, 0.456, 0.406], std [0.229, 0.224, 0.225]). For validation, we resize to 256 pixels on shorter side, center crop to $64 \times 64$, and apply the same normalization. We used 60,000 images with 50,000 images for train dataset and 10,000 images for validation dataset.

**NLP datasets** – We report performance on the GLUE development set following AdaLoRA (Zhang et al., 2023).

- **CoLA** (Warstadt et al., 2019): Judges if an English sentence is grammatically acceptable. (Train: 8.5k, Dev: 1k, Metric: Matthews Correlation Coefficient).

- **MNLI** (Williams et al., 2018): A 3-way classification task (entailment, neutral, contradiction) for sentence pairs across multiple genres. We use matched development set. (Train: 393k, Dev: 9.8k, Metric: Accuracy).

- **MRPC** (Dolan & Brockett, 2005): A binary classification task to determine if two sentences from online news are paraphrases. (Train: 3.7k, Dev: 408, Metric: Accuracy).

- **QNLI** (Rajpurkar et al., 2016): A binary classification task to identify if a context sentence contains the answer to a question. (Train: 105k, Dev: 5.4k, Metric: Accuracy).

- **QQP** (Iyer et al., 2017): A binary classification task to determine if two questions from Quora are semantically equivalent. (Train: 364k, Dev: 40k, Metric: Accuracy).

- **RTE** (Giampiccolo et al., 2007): A smaller, 2-way textual entailment classification task combining several datasets. (Train: 2.5k, Dev: 276, Metric: Accuracy).

- **SST-2** (Socher et al., 2013): A binary sentiment classification task on sentences from movie reviews. (Train: 67k, Dev: 872, Metric: Accuracy).

- **STS-B** (Cer et al., 2017): A regression task to predict a semantic similarity score (from 0 to 5) for sentence pairs. (Train: 5.7k, Dev: 1.5k, Metric: Pearson/Spearman Correlation).

**Vision Experimental Settings** – The Vision experiments detailed in Table 3 follow the configurations summarized in Table 6. We employed SGD optimizer with 0.9 momentum and conducted a grid search for learning rate, $I$, and $D$, executing each setting at least twice. The learning rate was tuned among 0.2, 0.1, 0.01. The $I$ and $D$ were first set to the midpoints of the high-noise and low-noise regimes, respectively. Then, each was finely tuned by grid search with a unit of 5. Table 6 presents the overall hyper-parameter settings we tuned.

**NLP Experimental Settings** – The NLP experiments presented in Table 4 are configured according to Table 7. We used AdamW optimizer with momentum 0.9, weight decay 1e-2, beta values (0.9, 0.999), sequence length 128, and 10% warm-up period of total steps. Through grid search performed at least twice per setting, we tuned learning rate among 1e-4, 5e-5, 2.5e-5, 1e-5, $D$ from 2, 3, and $\lambda$ among 0.5, 0.3, 0.1. Table 7 summarizes the highly tuned hyper-parameter settings.

**Experimental Settings of Rank Recovery Analysis** – The rank recovery experiments outlined in Table 5 follow the settings summarized in Table 8. We performed grid search for the algorithm-specific parameter, $\lambda$ (regularizer coefficient), from 1e-3, 5e-4, 1e-4, 5e-5, 1e-5.

Table 7: Hyper-parameter settings for experiments shown in Table 4. The $\lambda$ is the orthogonal regularizer coefficient in AdaLoRA (Zhang et al., 2023).

| Dataset | Batch Size | Learning Rate | Epochs ($E$) | LoRA rank | $\alpha$ | Algorithm-specific parameter | Deflate ($D$) |
|---------|-----------|---------------|--------------|-----------|----------|------------------------------|---------------|
| COLA | | $5e-5$ | 10 | | | $\lambda = 0.5$ | 5 |
| MNLI | | $1e-5$ | 5 | | | $\lambda = 0.1$ | |
| MRPC | | $1e-4$ | 5 | | | $\lambda = 0.1$ | 2 |
| QNLI | 16 | $1e-5$ | 5 | 16 | 16 | $\lambda = 0.1$ | |
| QQP | | $1e-5$ | 5 | | | $\lambda = 0.1$ | |
| RTE | | $5e-5$ | 10 | | | $\lambda = 0.1$ | 5 |
| SST-2 | | $1e-5$ | 5 | | | $\lambda = 0.1$ | 2 |
| SST-B | | $1e-4$ | 5 | | | $\lambda = 0.1$ | |

Table 8: Hyper-parameter settings for experiments shown in Table 5.

| Method | Batch Size | Learning Rate | Epochs ($E$) | LR Decay | Algorithm-specific parameter | Inflate/Deflate ($I, D$) | $\rho$ |
|--------|-----------|---------------|--------------|----------|------------------------------|--------------------------|--------|
| SO | | | | | $\lambda = 5e-5$ | | |
| DSO | 32 | 0.1 | 150 | 100, 130 | $\lambda = 5e-5$ | (55, 120) | 0.5 |
| SRIP$^+$ | | | | | $\lambda = 5e-4$ | | |

Table 9: Hyper-parameter settings for experiments shown in Table 10.

| Dataset | Batch Size | Learning Rate | Epochs ($E$) | LR Decay | $\phi$ | Inflate/Deflate ($I, D$) | $\rho$ |
|---------|-----------|---------------|--------------|----------|--------|--------------------------|--------|
| CIFAR-10 | 32 | 0.1 | 150 | 100, 130 | 0.1 | 92, 107 | 0.5 |
| | | | | | 0.3 | 75, 120 | |
| | | | | | 0.5 | 60, 135 | |
| | | | | | 0.7 | 45, 150 | |
| | | | | | 0.9 | 15, 150 | |
| CIFAR-100 | 32 | 0.1 | 200 | 150, 180 | 0.1 | 140, 160 | 0.5 |
| | | | | | 0.3 | 120, 180 | |
| | | | | | 0.5 | 100, 200 | |
| | | | | | 0.7 | 60, 200 | |
| | | | | | 0.9 | 20, 200 | |

**Experimental Settings of Ablation Study on Full-rank Epoch Budget** – We conducted an ablation study examining how the number of full-rank epochs affects model accuracy and computational cost. Table 9 summarizes experimental settings corresponding to Table 10. We follow popularly used hyperparameter settings (e.g., a batch size of 32 and a learning rate of 0.1, etc.) and adjusted $\phi$, $I$, and $D$. Given a fixed budget of $\phi E$ full-rank epochs, we allocate half to the high-noise regime and half to the low-noise regime.

## A.3 MODEL INFLATION / DEFLATION WITH VARIOUS DECOMPOSITION TECHNIQUES

Our dynamic-rank training method is compatible with various decomposition techniques. In the main manuscript, we described how to inflate and deflate models using SVD as an example. Here, we explain how model weights are reparameterized using Tucker and CP decompositions. Let $F$ denote the number of output channels, $C$ the number of input channels, $h$ the kernel height, $w$ the kernel width, and $k$ the reduced rank.

**Model Inflation with Tucker decomposition** – We define the model inflation process with Tucker decomposition (Kim et al., 2015) as follows. Given low-rank layer weights $\mathbf{A} \in \mathbb{R}^{1\times1\times C\times k}$, $\mathbf{Core} \in \mathbb{R}^{h\times w\times k\times k}$, $\mathbf{B} \in \mathbb{R}^{1\times1\times k\times F}$ and base parameter $\mathbf{W}_0 \in \mathbb{R}^{h\times w\times C\times F}$, the model is inflated such that $\mathbf{W} \leftarrow \mathbf{W}_0 + \mathbf{A}\mathbf{Core}\mathbf{B}$.

**Model Deflation with Tucker decomposition** – Given a model weight $\mathbf{W}$, the model is deflated by attaching a low-rank adaptor path next to the original weight such that $\mathbf{W}_f + \mathbf{A}\mathbf{Core}\mathbf{B} \leftarrow \mathbf{W}$, where $\mathbf{W}_f \in \mathbb{R}^{h\times w\times C\times F}$ is the given model weight matrix and $\mathbf{A} \in \mathbb{R}^{h\times w\times C\times F}$ and $\mathbf{A} \in \mathbb{R}^{1\times1\times C\times k}$, $\mathbf{Core} \in \mathbb{R}^{h\times w\times k\times k}$, and $\mathbf{B} \in \mathbb{R}^{1\times1\times k\times F}$ are low-rank model weights. The provided weight matrix is frozen as $\mathbf{W}_f$ and $\mathbf{A}$, $\mathbf{Core}$ and $\mathbf{B}$ are trained instead. One can initialize $\mathbf{A}$ and $\mathbf{B}$ using either random distributions or zero matrices.

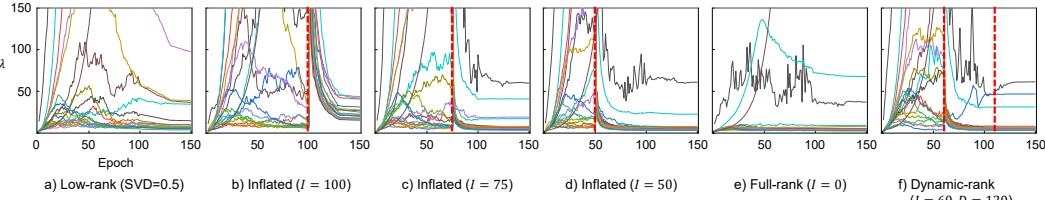

Figure 5: The singular value spectrum ratio $\lambda$ comparison. The red dotted lines indicate the epoch where the rank of model weights are adjusted.

**Model Inflation with CP decomposition** – We define the model inflation process with CP decomposition (Lebedev et al., 2014) as follows. Given low-rank layer weights $\mathbf{A} \in \mathbb{R}^{1\times1\times C\times k}$, $\mathbf{C_1} \in \mathbb{R}^{h\times1\times k\times k}$, $\mathbf{C_2} \in \mathbb{R}^{1\times w\times k\times k}$, $\mathbf{B} \in \mathbb{R}^{1\times1\times k\times F}$ and base parameter $\mathbf{W}_0 \in \mathbb{R}^{h\times w\times C\times F}$, the model is inflated such that $\mathbf{W} \leftarrow \mathbf{W}_0 + \mathbf{A}\mathbf{C_1}\mathbf{C_2}\mathbf{B}$.

**Model Deflation with CP decomposition** – Given a model weight $\mathbf{W}$, the model is deflated by attaching a low-rank adaptor path next to the original weight such that $\mathbf{W}_f + \mathbf{A}\mathbf{C_1}\mathbf{C_2}\mathbf{B} \leftarrow \mathbf{W}$, where $\mathbf{W}_f \in \mathbb{R}^{m\times n}$ is the given model weight matrix and $\mathbf{A} \in \mathbb{R}^{1\times1\times C\times k}$, $\mathbf{C_1} \in \mathbb{R}^{h\times1\times k\times k}$, $\mathbf{C_2} \in \mathbb{R}^{1\times w\times k\times k}$, and $\mathbf{B} \in \mathbb{R}^{1\times1\times k\times F}$ are low-rank model weights. The provided weight matrix is frozen as $\mathbf{W}_f$ and $\mathbf{A}$, $\mathbf{C_1}$, $\mathbf{C_2}$ and $\mathbf{B}$ are trained instead. One can initialize $\mathbf{A}$ and $\mathbf{B}$ using either random distributions or zero matrices.

## A.4 RANK ADJUSTMENT AN CONVOLUTION LAYERS

Let $F$ denote the number of output channels, $C$ the number of input channels, $h$ the kernel height, $w$ the kernel width, and $k$ the reduced rank. Note that we use a SVD-based approach as an example. Given low-rank convolution layer weights $\mathbf{A} \in \mathbb{R}^{h\times w\times C\times k}$, $\mathbf{B} \in \mathbb{R}^{1\times1\times k\times F}$ and base convolution layer weight $\mathbf{W_0} \in \mathbf{R}^{h\times w\times C\times F}$, convolution layer is inflated such that $\mathbf{W} \leftarrow \mathbf{W_0} + \mathbf{AB}$. Note that we assume $k < r \leq n$. If low-rank convolution follows the standard low-rank reparameterization method, the initial weight matrix $\mathbf{W_0}$ is set to zero matrix, i.e., $\mathbf{0}_{\mathbf{h}\times\mathbf{w}\times\mathbf{C}\times\mathbf{F}}$. Consequently, the maximum available rank is increased from $k$ to $n$ by training with inflated convolution layer. The deflation process follows the same steps, but in reverse order.

## A.5 SINGULAR VALUE SPECTRUM RATIO COMPARISON

To analyze the impact of interleaving full-rank epochs within low-rank training, we visualize the singular value spectrum ratio $\lambda^l, l \in [L]$ with various model inflation settings. Figure 5 presents the layer-wise $\lambda$ curves of five different inflation settings. As $I$ decreases, the full-rank epochs take up a large portion of the total epoch budget. We first observe that as $I$ decreases, the $\lambda^l$ values are more effectively suppressed, resulting in stable $\lambda^l$ curves across most layers. For example, in the full-rank curves shown in Figure 5.e), all curves remain below $\lambda^l < 20$. When the model is inflated too late (e.g., $I = 100$), the curves stay relatively high, indicating that the model weights have lost their rank. As shown in Figure 5.f), when $I$ is sufficiently small, the $\lambda^l$ values are significantly reduced in most layers. Even after the model rank is deflated at $D = 120$, the $\lambda^l$ values remain low until the end of training. This comparison provides clear insights into how to dynamically adjust the model rank during training to maximize the rank of the model weights.

Another intriguing observation is that the $\lambda$ values at a few layers remain high regardless of rank adjustment. Notably, these layers consistently include the first and last layers across all experiments. Since their inputs or outputs remain fixed during training, their weights may rapidly fit to data patterns. If their rank can be suppressed, the model's overall capacity may be more effectively utilized, potentially achieving higher accuracy within the same epoch budget. We consider this an interesting direction for future research.

Table 10: Ablation study on how $\phi = (D - I)/E$ affects the performance of dynamic-rank training. Low-Rank SVD is used for all experiments. When $\phi = 0.0$, it becomes the conventional low-rank training. In contrast, when $\phi = 1.0$, it becomes full-rank training. The $I$ and $D$ are set to perform the same number of full-rank epochs in the high-noise and low-noise regimes.

| Setting | Rank Ratio $\rho$ | Comp. | CIFAR-10 | CIFAR-100 |
|---------|-------------------|-------|----------|-----------|
| $\phi = 0.1$ |       | 0.36 | $90.54 \pm 0.3\%$ | $76.29 \pm 0.1\%$ |
| $\phi = 0.3$ |       | 0.51 | $91.43 \pm 0.1\%$ | $77.03 \pm 0.4\%$ |
| $\phi = 0.5$ | 0.25  | 0.64 | $92.01 \pm 0.3\%$ | $78.61 \pm 0.2\%$ |
| $\phi = 0.7$ |       | 0.79 | $92.08 \pm 0.4\%$ | $78.75 \pm 0.1\%$ |
| $\phi = 0.9$ |       | 0.93 | $92.14 \pm 0.2\%$ | $78.63 \pm 0.3\%$ |
| $\phi = 0.1$ |       | 0.61 | $91.28 \pm 0.1\%$ | $76.84 \pm 0.1\%$ |
| $\phi = 0.3$ |       | 0.69 | $91.57 \pm 0.1\%$ | $77.51 \pm 0.2\%$ |
| $\phi = 0.5$ | 0.5   | 0.78 | $92.11 \pm 0.2\%$ | $78.41 \pm 0.3\%$ |
| $\phi = 0.7$ |       | 0.87 | $92.15 \pm 0.2\%$ | $78.39 \pm 0.1\%$ |
| $\phi = 0.9$ |       | 0.96 | $92.11 \pm 0.1\%$ | $78.47 \pm 0.3\%$ |

### A.6 ADDITIONAL ABLATION STUDY

In our empirical study, we found that dynamic-rank training performs well when $D$ and $I$ are set to allocate a similar number of full-rank epochs to both the high-noise and low-noise regimes. We now conduct a simple ablation study to examine how the number of full-rank epochs affects model accuracy. Table 10 presents the results. SVD-based low-rank training is evaluated on the CIFAR-10 and CIFAR-100 benchmarks. For convenience, we define $\phi = (D - I)/E$ as the ratio of full-rank epochs to the total training budget. When $\phi = 0.0$, it corresponds to conventional low-rank training; when $\phi = 1.0$, it becomes full-rank training. On both benchmarks, accuracy starts to drop when $\phi$ goes below 0.5, particularly in the range $\phi \in [0.3, 0.5]$. We therefore recommend starting with $\phi = 0.5$ and adjusting downward as needed.

### A.7 POTENTIAL LIMITATIONS AND FUTURE WORK

**Potential Limitations** – Our proposed method has a relatively more expensive computational cost compared to the conventional low-rank training methods. Since the number of trainable parameters increases during $D - I$ full-rank epochs, the overall cost is increased as shown in *Comp.* column in Table 3. However, the cost still remains substantially lower than that of the full-rank training, and we argue that the dynamic-rank training is a practical option for large-scale deep learning applications.

In addition, the extra hyperparameters, $I$ and $D$, can introduce a non-trivial tuning overhead. However, Section 4.2 provides useful guidance on how to select good values for $I$ and $D$ based on learning rate decay schedules. In our empirical study, we find that $I$ and $D$ can be tuned easily by following our suggestions, leading to substantial accuracy improvement while maintaining low computational cost.

**Future Work** – We plan to extend our dynamic-rank training research to automate the inflation and deflation steps. Our study reveals that the inflation should be located at the end of the high-noise regime and the deflation should be as early as possible in the low-noise regime. If the noise scale can be quantified at run time, rather than explicitly mapping it to the learning rate decay schedule, appropriate timings for increasing and decreasing the model rank can be identified during training. We believe this will make the dynamic-rank training framework significantly more practical. Moreover, we consider harmonizing the dynamic-rank training framework with other existing compute-efficient training strategies as a promising direction for future work. Given the ever-increasing model sizes in deep learning applications (e.g., LLMs), developing neural network training strategies that balance accuracy and efficiency is a crucial research direction.

