# OpenReview forum: "Dynamic Rank Adjustment for Accurate and Efficient Neural Network Training"
_ICLR.cc/2026/Conference — ICLR 2026 Conference Withdrawn Submission_

### Official Review · Reviewer_ofNH · 2025-10-21

**Soundness:** 2
**Presentation:** 2
**Contribution:** 1
**Rating:** 2
**Confidence:** 4

**Summary:**

In this work, the authors propose a new framework to avoid the reduced capacity of low-rank models while pretraining. In particular, the authors propose to alternate phases of low-rank adaptation and phases of rank increase in which the model can increase its capacity. The proposed framework is evaluated on a variety of different benchmarks, both for vision and language tasks.

**Strengths:**

The paper is overall well-written and organized clearly. The investigated problem is very relevant in the contect of stable pretraining and fine-tuning.

**Weaknesses:**

1. I personally fail to see the point of Proposition 2: while it is true that the right-hand side increases as a function of the learning rate, the bound on $d_t$ given by
$$
d_t \leq ||\nabla f(W_t)||_F(1+ O(\eta)) + O(\eta^2) \to ||\nabla f(W_t)||_F, \quad \eta \to 0
$$
Therefore, the tightest bound is obtained in the limit, but it simply says that the relative error
$$
\frac{d_t}{||\nabla f(W_t) ||} \leq 1,
$$
in the limit $\eta \to 0$. For this reason, the bound is fully vacuous, and therefore I fail to see the point of this result.

2. Proposition 1 is a very classical result, I would include it in the form of a remark or observation with reference.

3. [1] is an important piece of literature that the authors failed to discuss and compare with, which is essentially a similar philosophy with the only difference that the rank of the overall weights can increase at each optimization step (by a small amount typically since updates are low-rank).


[1] W. Xia et al., Chain of LoRA: Efficient Fine-tuning of Language Models via Residual Learning, ICML 2024.

**Questions:**

1. It is not very clear how the rank inflation in Algorithm 1 is performed, by performing a couple of optimization steps on $W$?

I would also appreciate having a discussion with the authors concerning the weaknesses.

---

### Official Review · Reviewer_dLgF · 2025-10-31

**Soundness:** 2
**Presentation:** 2
**Contribution:** 2
**Rating:** 2
**Confidence:** 5

**Summary:**

This paper proposes a dynamic-rank training framework to address a key limitation of low-rank training, the permanent loss of representational capacity and rank collapse.

**Strengths:**

+ The proposed dynamic-rank training can somewhat increase the model capacity in the fine-tuning process while preserving higher performance.
+ This paper provides a theoretical analysis of the proposed dynamic-rank training.
+ The proposed method performs well on multiple datasets compared to existing low-ranking training approaches.

**Weaknesses:**

- Limited novelty. Although the paper adopts a dynamic-rank strategy, the approach essentially remains a variant of standard low-rank training. Dynamically adjusting the rank appears to be more of a training trick than a genuine research innovation. Moreover, the observation that higher ranks yield better performance is a well-known and intuitive fact rather than a novel insight.

- Questionable practicality. The proposed method increases training cost due to periodic rank adjustments, which require significantly more memory and time compared to fixed-rank approaches. Additionally, designing the rank-scheduling policy adds extra hyperparameters and implementation complexity. It is unclear how easily the proposed dynamic schedule generalizes across architectures and optimizers. The method may require task-specific tuning of schedule parameters, limiting its practical adoption. According to Figure 2, the approach requires full-rank phases, which makes it conceptually similar to fine-tuning the original full model, reducing its practical advantage.

- Lack of runtime and memory analysis. Although the method achieves accuracy improvements, the paper does not report actual runtime or memory usage. In practical scenarios, efficiency, both in terms of memory footprint and fine-tuning time, is often more critical than small accuracy gains.

**Questions:**

See Weaknesses.

---

### Official Review · Reviewer_VwGw · 2025-11-01

**Soundness:** 2
**Presentation:** 3
**Contribution:** 2
**Rating:** 4
**Confidence:** 4

**Summary:**

This paper introduces Dynamic Rank Training (DRT), a framework that strategically interleaves full-rank "restorative" phases within low-rank training to mitigate effective rank decline. The key insight is to schedule these phases in alignment with the learning rate's noise regime by placing them at the transition between high and low noise to maximally recover model capacity. Extensive evaluations across vision and NLP benchmarks demonstrate that DRT achieves accuracy comparable to full-rank training while retaining the computational benefits of low-rank methods, and proves compatible with various decomposition techniques and orthogonal regularization methods.

**Strengths:**

1. The paper is well organized and the writing, figures, and algorithmic pseudocode are easy to follow.

2. Evaluations on diverse benchmarks are thorough and show consistent performance improvements with moderate computational overhead.

**Weaknesses:**

1. Limited Theoretical Depth. The theoretical analysis remains heuristic, and does not provide rigorous convergence guarantees or formal explanation on how the scheduling principle optimizes rank recovery.

i) Proposition 1 provides a bound on the rank of the reconstructed matrix that is derived under an idealized assumption. The low-rank component is supposed to perfectly cancel the base weights, which may not reflect the complex, stochastic optimization dynamics in practice.

ii) Similarly, Proposition 2 shows that the update gap is constrained by a function of the learning rate. However,  Proposition 2 cannot formally guarantee that the proposed scheduling strategy minimizes the cumulative gap or for maximizing rank recovery.

2. Sensitivity to Scheduling Hyperparameters. DRT introduces critical new hyperparameters of inflation (I) and deflation (D) epochs that could rely on the dataset, and could be sensitive to the inflation/deflation schedule, as shown in the appendix where different I/D are used for CIFAR-10 and CIFAR-100. It is non-trivial to identify the optimal schedule for the dataset-specific tuning and could cause computationally expensive search process. High-level scheduling principles are provided in Section 4.2 but are still vague guidelines rather than an automated or theoretically grounded procedure. Theoretical or empirical justification for determining a near-optimal schedule in a model-agnostic way is lacking, especially for billion-parameter architectures. Sensitivity to scheduling hyperparameters hampers the framework from a robust and general-purpose efficiency solution.

3. Insufficient Empirical Validation on Scalability and Practical Efficiency. The empirical evaluation does not fully support the generalizability of the proposed framework, as it lacks testing on large-scale foundation models where low-rank efficiency is most critical. Moreover, the computational analysis relies on an averaged parameter count, but does not provide peak memory usage and actual time overhead during full-rank phases that are usually leveraged to evaluate the practical utility.

**Questions:**

Please refer to the section of weaknesses.

---

### Note · Authors · 2025-11-13

**Comment:**

Review quality is very poor, and we believe there will be no constructive discussions available. So withdraw our manuscript.

**Withdrawal Confirmation:**

I have read and agree with the venue's withdrawal policy on behalf of myself and my co-authors.